# PanoLAM : Large Avatar Model for Gaussian Full-Head Synthesis from One-shot Unposed Image

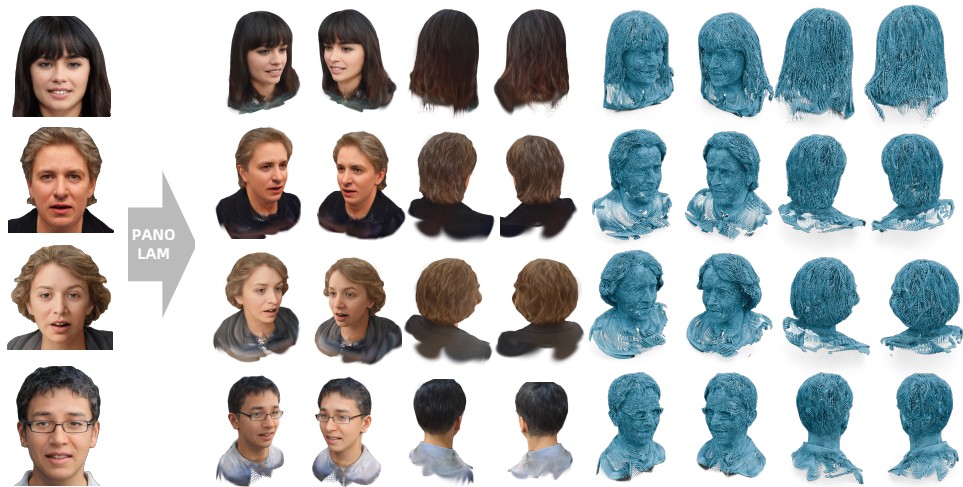

Figure 1: PanoLAM creates high-fidelity Gaussian full-heads with one-shot unposed images in seconds.

## Abstract

We present a feed-forward framework for Gaussian full-head synthesis from a single unposed image. Unlike previous work that relies on time-consuming GAN inversion and test-time optimization, our framework can reconstruct the Gaussian full-head model given a single unposed image in a single forward pass. This enables fast reconstruction and rendering during inference. To mitigate the lack of large-scale 3D head assets, we propose a large-scale synthetic dataset from trained 3D GANs and train our framework using only synthetic data. For efficient high-fidelity generation, we introduce a coarse-to-fine Gaussian head generation pipeline, where sparse points from the FLAME model interact with the image features by transformer blocks for feature extraction and coarse shape reconstruction, which are then densified for high-fidelity reconstruction. To fully leverage the prior knowledge residing in pretrained 3D GANs for effective reconstruction, we propose a dual-branch framework that effectively aggregates the structured spherical triplane feature and unstructured point-based features for more effective Gaussian head reconstruction. Experimental results show the effectiveness of our framework towards existing work.

## 1 Introduction

Reconstructing head avatars from a single image is an important area of research within computer graphics and computer vision. The capability to generate high-fidelity head avatars opens up new possibilities in areas such as 3D telepresence, video conferencing, filmmaking, gaming, and augmented/virtual reality (AR/VR). However, efficiently creating digital head avatars that are high-fidelity and efficient in rendering is challenging, and various studies have tried to resolve this problem.

One line of work uses 3D Generative Adversarial Networks (3D-GANs) (Chan et al., 2022a; An et al., 2023) for the unconditional generation of 3D head avatars, which learn 3D generation models from 2D

images paired with camera poses. Unlike 2D-GANs (Karras et al., 2019) for single-view generation, these approaches can generate face images with control over viewing angles. The pioneering work EG3D (Chan et al., 2022a) introduces a triplane representation in Cartesian space to enable novel view synthesis after generation. The following works (An et al., 2023; Li et al., 2024) allow a wider range of viewing angles from the generated NeRF space and make 360° images synthesis possible. However, these GAN-based frameworks require time- and computation-consuming GAN-inversion (Ko et al., 2023) or test-time optimization (Roich et al., 2023) for image-conditioned generation during inference. Furthermore, the accurate camera pose of the input image is required for joint optimization of these inversion approaches. Rodin (Wang et al., 2023a) and its follow-up (Zhang et al., 2024a) instead utilize diffusion models to generate triplanes of the head avatars. However, the multi-step diffusion process during inference still requires minutes of optimization for each case. Moreover, the costly ray marching hinders the rendering resolution of these NeRF-based frameworks. Though 2D super-resolution upsampling is proposed to improve the rendering efficiency and quality, it compromises 3D consistency.

In this work, we propose a feed-forward framework to generate a full-head Gaussian avatar given a single *unposed* image. This enables fast reconstruction and rendering of a 3D avatar during inference. However, this task presents certain challenges. **1)** The first hurdle lies in the lack of large-scale 3D head datasets for network training. To tackle this problem, we propose to leverage the prior knowledge in trained 3D GANs and sample a large-scale, diverse dataset from EG3D (Chan et al., 2022a) and SphereHead (Li et al., 2024). We label and remove invalid cases manually to improve the quality of the dataset. **2)** The second challenge involves how to efficiently and effectively reconstruct the high-fidelity Gaussian head from a single unposed image. To overcome this, we propose two main designs for our framework. **i)** Firstly, to improve the network efficiency, we propose a coarse-to-fine reconstruction mechanism for high-fidelity Gaussian full-head reconstruction. Specifically, sparse points extract rich image features, craft the coarse shape, and are then upsampled for high-fidelity dense Gaussian head reconstruction. By leveraging the topology prior in the FLAME model, we also introduce an efficient network-free upsampling mechanism for features and point clouds densification. **ii)** Secondly, to improve the effectiveness of the reconstruction process, we propose a dual-branch framework that hybridizes the unstructured point features and the structured spherical triplane features for the reconstruction of a Gaussian head. To distill the spherical triplane knowledge from the pretrained 3D GAN, we analyze its original feature aggregation manner and propose an efficient mechanism to effectively aggregate multi-layer features from the spherical triplane.

The principal contributions of our work can be summarized as follows:

- We propose PanoLAM, a large avatar model for *Gaussian full-head* reconstruction from *a single unposed* image.

- We propose a coarse-to-fine reconstruction mechanism for efficient and high-fidelity Gaussian full-head reconstruction. Built upon the topology prior in the FLAME model, we introduce an efficient network-free upsampling mechanism for points and features densification.

- We propose a dual-branch framework that hybridizes the representation of point and the spherical triplane prior from 3D GAN. An efficient aggregation mechanism is also proposed to effectively extract multi-layer features from the triplane.

- We propose a large-scale, diverse synthesis dataset for 3D avatar reconstruction and generation. To our knowledge, ours is the largest and most diverse 3D head avatar dataset.

## 2 RELATED WORK

**3D Head Modeling Models.** Various 3D representations have been applied for 3D head modeling, including mesh, Neural Radiance Fields (NeRFs) (Mildenhall et al., 2020), Signed Distance Functions (SDFs), and Gaussian Splatting (Kerbl et al., 2023). Mesh-based modelings (Blanz & Vetter, 1999; Paysan et al., 2009; Li et al., 2017) are dedicated to representing 3D human heads with parametric textured meshes and deform the mesh model to improve the topology (Khakhulin et al., 2022; Liao et al., 2025). NeRF-based works (Yu et al., 2023; Ma et al., 2023; Li et al., 2023a; Ki et al., 2024; Bai et al., 2023a; Park et al., 2021a; Zielonka et al., 2023; Zhang et al., 2024b; Zhao et al., 2024; Bai et al., 2023b; Guo et al., 2021; Gao et al., 2022; Park et al., 2021b; Athar et al., 2022; Hong et al., 2021; Tretschk et al., 2021; Gafni et al., 2021) utilize NeRF for 3D head modeling and get more realistic

rendering. To improve geometry quality, SDF-based methods (Zheng et al., 2025; 2022; Zanfir et al., 2024; Canela et al., 2024) are also introduced, where each 3D position is assigned with the signed distance to the nearest surface. For more efficient rendering, the point-based (Zheng et al., 2023) and Gaussian Splatting-based (Qian et al., 2024; Xu et al., 2024; Saito et al., 2024; Ma et al., 2024; Dhamo et al., 2024; Chen et al., 2024) frameworks are also proposed. However, these frameworks usually require minutes to hours of optimization from videos or multiview images with estimated camera pose for each person before usage, limiting their capability of scaling up and applications that require fast reconstruction. Instead, our work introduces a framework that can generate a Gaussian *full-head* avatar from *a single unposed image* in *a single forward pass* within *seconds*.

**3D Avatar Generative Models.** One line of generative models utilizes 3D-aware Generative Adversarial Networks (GANs) (Chan et al., 2022a) to synthesize view-consistent images. Early approaches (Nguyen-Phuoc et al., 2020; 2019; Gadelha et al., 2017; Szabó et al., 2019; Shi et al., 2021; Liao et al., 2020) employ explicit 3D representations like meshes and voxel grids, while more recent studies (Chan et al., 2022a; An et al., 2023) utilize implicit representations for better image quality. EG3D (Chan et al., 2022a) and its follow-ups (An et al., 2023; Li et al., 2024) utilize 3D triplane representation with GANs to generate heads that are capable of novel view synthesis. However, these methods are mainly designed for unconditional generation, and require time- and computation-consuming techniques like GAN inversion (Sun et al., 2022; Chan et al., 2022b; Ko et al., 2023) and test-time optimization (Roich et al., 2023) for image-conditioned generation. The slow optimization procedure during inference and the sacrifice in multi-view consistency hinder their real-world application. Trevithick et al. (2023); Bhattarai et al. (2024); Yuan et al. (2023); Chen et al. (2023) distill an encoding network from EG3D for efficient image-to-triplane generation, but can only generate views near the front face. Instead of using GAN-based frameworks, more recent works (Wang et al., 2023a; Zhang et al., 2024a) utilize diffusion models to generate triplanes of head avatars. However, the multi-step diffusion process is still slow and computation-intensive during inference. Several works (Chu et al., 2024; Abdal et al., 2024; Sun et al., 2023; Kirschstein et al., 2025; Deng et al., 2024; Li et al., 2023b) also introduce generalizable animatable avatar generation, but cannot reconstruct full heads. Moreover, due to the slow rendering speed of NeRF, these approaches require 2D super-resolution networks to enhance image detail, causing view inconsistencies. To enhance rendering quality and speed, recent works (He et al., 2025; Chu & Harada, 2024; Kirschstein et al., 2024; Tang et al., 2024) also utilize Gaussian Splatting (Kerbl et al., 2023) as the 3D representation for generation. LAM (He et al., 2025) introduces a feedforward framework for single-shot animatable Gaussian head generation that can be animated and rendered in real-time on various platforms. However, it requires estimation of FLAME parameters for reconstruction, which is time-consuming, and the generated head can only be rendered from limited viewing angles due to the limited viewing angles in the training 2D videos. Instead, this work can reconstruct the 3D *Gaussian full-head* given a single *unposed* image *in a single forward pass*, thanks to our large-scale 3D training datasets and our coarse-to-fine and dual-branch network design.

## 3 METHODOLOGY

### 3.1 OVERVIEW

As shown in Fig. 2, given an unposed reference image, our framework generates the Gaussian full-head in an efficient coarse-to-fine and joint point-triplane manner, where dense point cloud and point features are densified from sparse ones and fused with the spherical triplane features for dense Gaussian head regression. As shown in the upper left part of Fig. 2, in the coarse stage of the point branch, sparse vertices from the FLAME vertices interact with the extracted multi-level image features by stacked cross-attention modules for feature extraction and coarse Gaussian point reconstruction. The dense reconstruction stage leverages barycentric weights from FLAME subdivision to densify the coarse Gaussian point position and features for dense Gaussian points regression. To fully leverage the prior knowledge residing in the spherical triplane of 3D GAN, as shown in the bottom branch of Fig. 2, a spherical triplane branch is also introduced, and a novel efficient multi-layer aggregation mechanism is proposed to aggregate the structured features from the spherical triplane using the densified point cloud, which are then concatenated with the unstructured point features for high-fidelity Gaussian head reconstruction. To optimize the spherical-triplane branch and distill the prior knowledge, in the bottom right of the framework, a frozen SphereHead decoder is utilized to decode the feature to images for supervision during training.

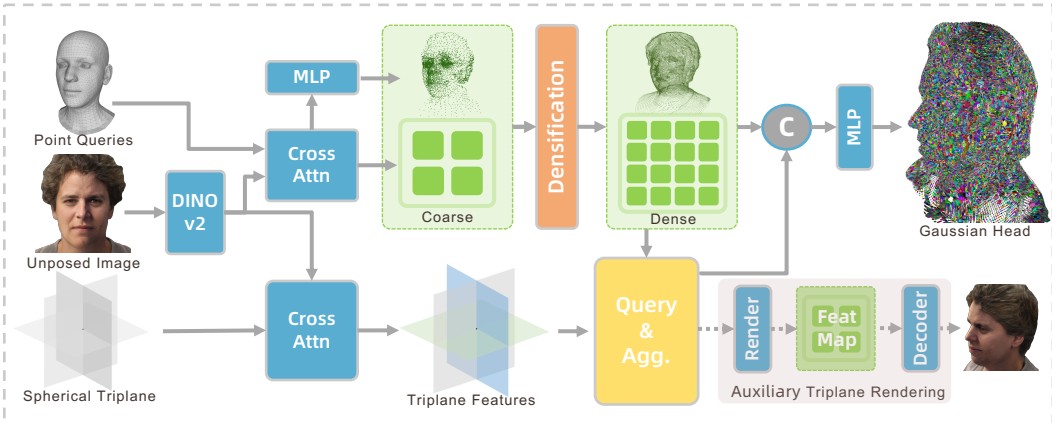

Figure 2: **Overall Framework.** Given an unposed head image as input, PanoLAM involves two branches to achieve single-pass 3D Gaussian head reconstruction: a point-based transformer for *coarse-to-fine* point shape reconstruction and point features extraction, and a spherical triplane transformer to distill prior knowledge from 3D GAN. Features from the two branches are concatenated for high-fidelity Gaussian head regression.

## 3.2 COARSE-TO-FINE GAUSSIAN HEAD AVATAR GENERATION

The number of Gaussian points is one key factor that can affect the fidelity of the reconstructed Gaussian head, especially for the head avatar that contains sharp details like hair strands, mustaches, and wrinkles. The total number of Gaussian points should be large enough to produce these details. However, the large number of Gaussian points is hard to optimize during training, especially in our feed-forward pipeline, where the iterative densification operation in the traditional 3D Gaussian Splatting training pipeline is not applicable. To resolve this problem, unlike previous works (He et al., 2025; Zou et al., 2024) that directly regresses the dense point cloud, we introduce a coarse-to-fine mechanism. Specifically, sparse Gaussian points are first reconstructed for coarse shape reconstruction and then densified to dense Gaussians for high-fidelity details crafting. There are two advantages in the proposed design. Firstly, the sparse number of points enables efficient cross-attention with the image features for textures and shape features extraction, saving large computation and memory costs compared with a large number of points. Secondly, the sparse point can be well supervised and optimized for large deformation to carve the shape of the head, especially in regions that require large deformation from the original FLAME model, e.g., long hairs. Dense Gaussian points upsampled from these correctly deformed sparse points can focus more on texture details crafting, which are much easier to optimize.

**Coarse Gaussian Head Generation**   To fully leverage the head shape prior that resides in the FLAME model, we initialize the coarse Gaussian point with sparse vertices (5,023 points) on the neutral FLAME model in the canonical space. Our target is to regress the deformation offset for shape refinement of regions that FLAME cannot model, e.g., long hairs, glasses, and caps. To supervise the shape deformation with RGB input images, Gaussian attributes for each point are also regressed for differentiable Gaussian Splatting. To get started, we assign positional embedding to each point and utilize learnable multi-layer perceptron (MLP) modules to project the channel of features into the token channel $C_t$ of transformers as: $F_{P_0} = MLP(\gamma(V))$, where $V$ is the spatial position of each vertice and $\gamma$ the L_frequency sinusoidal encoding as in NeRF. To extract the texture and shape information in the given unposed image, we utilize the pre-trained DinoV2 (Oquab et al., 2024) for feature extraction. Inspired by (Ranftl et al., 2021; He et al., 2025), we fuse features derived from both shallow and deep layers to obtain both local and global image features $F_I$. Specifically, an MLP fuses the $\{5, 12, 18, 24\}$ layers of features in DinoV2 into $C_t$ token channels of features as:

$$F_I = MLP(\mathcal{C}(F_{D_5}, F_{D_{12}}, F_{D_{18}}, F_{D_{24}}), \tag{1}$$

where $F_{D_i}$ is the $i_{th}$ layer of DinoV2 features and $\mathcal{C}$ is the concatenation operation.

Given mapped point features $F_P$ and extracted image features $F_I$, we utilize $L_{\mathcal{A}}$ layers of stacked cross-attention modules $\mathcal{A} = \{\mathcal{A}_i\}_{i=1}^{L_{\mathcal{A}}}$ from Transformer (Vaswani et al., 2017) for feature extraction from image to point cloud as follows:

$$F_{P_i} = \mathcal{A}_i(F_{P_{i-1}}, F_I), \tag{2}$$

where $F_{P_i}$ is the $i_{th}$ layers of point features in the stacked cross-attention layers.

Following the feature extraction process, each point retains its distinct features. Using these features, we develop decoding headers $\mathcal{D}$, composed of multilayer perceptrons (MLPs), to predict the deformation offset $O_k \in \mathbb{R}^3$ to refine the individual's detailed shape. Gaussian attributes for each point are also regressed for rendering, including color $c_k \in \mathbb{R}^3$, opacity $o_k \in \mathbb{R}$, scale factors $s_k \in \mathbb{R}^3$, and rotation $R_k \in SO(3)$. This decoding process is as follows:

$$\{c_k, o_k, s_k, R_k, O_k\}_{k=1}^{M_C} = \mathcal{D}(F_{P_k}), \tag{3}$$

where $M_C = 5023$ represents the total number of Gaussians in the coarse reconstruction stage, and $F_{P_k}$ denotes the extracted point feature for point $P_k$. Although predicting deformation offsets for each point can enhance the shape, freely moving these points may also negatively impact reconstruction results. We therefore restrict the range of deformation offset to be within $[-\epsilon_{O_C}, \epsilon_{O_C}]$.

**Efficient Point Cloud Densification**    After the reconstruction of the coarse Gaussian head, we obtain a sparse point cloud that describes the shape of the target head avatar in the image, each point is also attached with rich features extracted from the image. We then densify these points and features for high-fidelity dense Gaussian head reconstruction. Unlike previous methods (Li et al., 2021; Yu et al., 2018) that require a network trained to upsample sparse points, we propose to leverage the topology in the FLAME mesh for efficient densification. Specifically, the proposed strategy utilizes the barycentric coordinates from mesh subdivision to densify the point cloud and feature. Our key observation is that after the deformation in the coarse reconstruction stage, the topology between each point mostly remains as the original FLAME vertices. Thereafter, we can precompute the barycentric coordinates for point cloud densification utilizing mesh subdivision on FLAME. Specifically, given the original FLAME model, we perform subdivision for faces that are larger than a threshold $\lambda_{area}$, the position of newly added vertices $v'$ can be computed from the three vertices of the original face with barycentric coordinates as: $v' = \sum_{i=1}^{3}(w_i \cdot v_i)$, where $w_i$ and $v_i$ are the barycentric weight and vertex coordinate of the original face, respectively. After the deformation in the coarse reconstruction stage, each vertex moves to a new position w.r.t the predicted offset as $p_i = v_i + O_i$. Then new added point $p'$ with its attached point features $p'_F$ can be interpolated with barycentric weights as: $p' = \sum_{i=1}^{3}(w_i \cdot p_i)$, and $p'_F = \sum_{i=1}^{3}(w_i \cdot p_{F_i})$, where $p_{F_i}$ is the extracted features from the last cross-attention blocks in Equ. (2). In this way, we can efficiently densify the point and features.

**Dense Gaussian Head Generation.**    After getting the dense point position with dense point features, we can utilize them for dense Gaussian head regression. Rather than using only unstructured point features for prediction, we also aggregate multi-layer of features from the structured spherical triplane for more effective Gaussian attributes regression, which we will discuss in Sec. 3.3. Denote $F_{P_i}$ the $i_{th}$ point feature and $F_{T_i}$ the aggregated spherical triplane feature for point $P_i$, the dense Gaussian head attributes and a small deformation residual can be regressed similarly to the coarse one as:

$$\{c_i, o_i, s_i, R_i, O_i\}_{i=1}^{M_D} = \mathcal{D}(\mathcal{C}(F_{P_i}, F_{T_i})), \tag{4}$$

where $\mathcal{C}$ denotes the concatenation operation, $M_D$ the number of dense Gaussian points, and $O_i$ the deformation residual restricted within $[-\epsilon_{O_D}, \epsilon_{O_D}]$.

### 3.3 SPHERICAL TRIPLANE PRIOR KNOWLEDGE DISTILLATION

In the previous section, we obtain the dense point cloud with its attached point features extracted from the unposed image. To better leverage the prior knowledge within SphereHead's framework, we introduce a spherical triplane branch to distill the knowledge from a pretrained model.

**Preliminary: The Spherical Triplane representation**    from SphereHead Li et al. (2024) combines the three triplanes in the Cartesian coordinate system with another three spherical planes in the spherical coordinate system, where features are queried with projection operations in the respective coordinate systems. This representation alleviates the feature entanglement issue in regular triplanes.

In the spherical triplane branch, we initialize the $H_T \times W_T \times 6 \times C_t$ spherical triplane features with learnable query features, which are then flattened into token features $F_T \in \mathbb{R}^{N_T \times C_t}$, where $N_T = H_T \times W_T \times 6$. These learnable features then interact with the image features $F_I$ with stacked

cross-attention blocks $\mathcal{A}_\mathcal{T} = \{\mathcal{A}_{\mathcal{T}i}\}_{i=1}^{L_{\mathcal{A}_\mathcal{T}}}$ similar to the learnable point features as:

$$F_{T_i} = \mathcal{A}_{\mathcal{T}i}(F_{T_{i-1}}, F_I), \tag{5}$$

where $F_{T_i}$ is the $i_{th}$ layer of spherical triplane features in the stacked cross-attention layers. The final output features are then projected by MLPs to the same channel as the spherical triplane in SphereHead.

We then need to query the feature from the spherical triplane to enhance the point features. One vanilla way is to utilize the point-based query strategy that fetches each point's spherical-triplane feature by projection. However, we find that our point clouds are distributed on the shape surface similar to a mesh, and such a single-layer query strategy cannot fully aggregate the corresponding features of each point from the spherical triplane. That's because the triplane designed for NeRF-based rendering requires ray marching to aggregate multiple points(/layers) of features for rendering. Using a single point cloud on the shape surface cannot fully fetch valuable features from the spherical triplane. While ray marching is computationally consuming and inefficient, we pro-

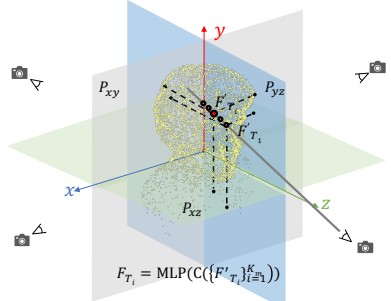

Figure 3: The proposed feature aggregation mechanism samples and aggregates multi-layer features from the spherical triplane for each point.

pose to sample multiple layers of points from the single-layer point cloud for feature aggregation. As is shown in Fig. 3, we sample 4 virtual cameras that can capture the front, left, right, and back sides of the head. For each camera, we cast a ray from the camera center to each point $P_i$, then $K_m$ number of points $P' = \{P'_i\}_{i=1}^{K_m}$ are sampled along the ray near the point. Each point then queries features from the spherical triplane by projection. We then concatenate and aggregate these queried features with MLPs to obtain the final spherical triplane features as:

$$F_{T_i} = MLP(\mathcal{C}(\{F'_{T_i}\}_{i=1}^{K_m})), \tag{6}$$

where $F'_{T_i}$ is the spherical triplane feature queried by sampled point $P'_i$ ; $\mathcal{C}$ is the concatenation operation; and $F_{T_i}$ is the final aggregated feature for point $P_i$. In this way, each point gets its aggregated spherical triplane feature. These aggregated spherical triplane features can be concatenated with the point feature for dense Gaussian attribute regression as in Formula (4). To supervise this branch in distilling knowledge from SphereHead, we add a frozen decoder from SphereHead to decode the feature into the target image, supervised by ground truth images during training, as shown in the bottom right part of Fig 2.

## 3.4 LARGE-SCALE TRAINING DATA GENERATION

The scale of the training dataset is a key factor in enabling the generalizability of a trained model. However, obtaining a large dataset with diverse head assets is challenging. Though various datasets (Yin et al., 2006; Zhang et al., 2014b; Hu et al., 2007; Savran et al., 2008; Cao et al., 2014; Cheng et al., 2018; Zhang et al., 2014a; Dai et al., 2020; Kirschstein et al., 2023; Cosker et al., 2011; Zhu et al., 2023; Wang et al., 2022; Pan et al., 2023; He et al., 2024; Martinez

Table 1: Comparisons of different 3D avatar datasets. $^*$ denotes 3D head with rough captured 3D hair shape.

| Dataset | Sub. | Range | Hair | Dataset | Sub. | Range | Hair |
|---|---|---|---|---|---|---|---|
| BU-3DFE | 100 | front | ✗ | HeadSpace | 1519 | 270° | ✗ |
| BU-4DFE | 101 | front | ✓ | FaceScape | 938 | 360° | ✗ |
| BJUT-3D | 500 | front | ✗ | AVA-256 | 256 | 360° | ✓ |
| Bosphorus | 105 | front | ✗ | FaceVerse$^*$ | 128 | 360° | ✓ |
| FaceWarhouse | 150 | front | ✓ | RenderMe360$^*$ | 500 | 360° | ✓ |
| 4DFAB | 180 | front | ✗ | SynHead100 | 100 | 360° | ✓ |
| BP4D-S | 41 | front | ✓ | **Ours-Front** | 48,135 | 72° | ✓ |
| Nersemble | 222 | front | ✓ | **Ours-360** | 117,186 | 360° | ✓ |

et al., 2024) have been proposed for 3D head modeling, as shown in Table 1, they have limited diversity of subjects, which hinders the generalization capability of learning-based models. Inspired by previous methods (Peebles et al., 2022; Cai et al., 2024; He et al., 2022; Wood et al., 2021) in various fields that model trained from synthesis data can generalize to real-world scenarios, we leverage pretrained priors from 3D GANs to generate a large-scale, diverse dataset for training. Specifically, we leverage two prior models for data generation, EG3D for multiview front face images synthesis, and SphereHead for 360° view images generation. Our observation is that EG3D generates a different distribution of images compared to SphereHead (e.g., diverse hairstyles and caps appeared in samples), while SphereHead can generate 360° images. The combination of two model spaces enhances the diversity of the generated images and improves the generalizability of learning-based

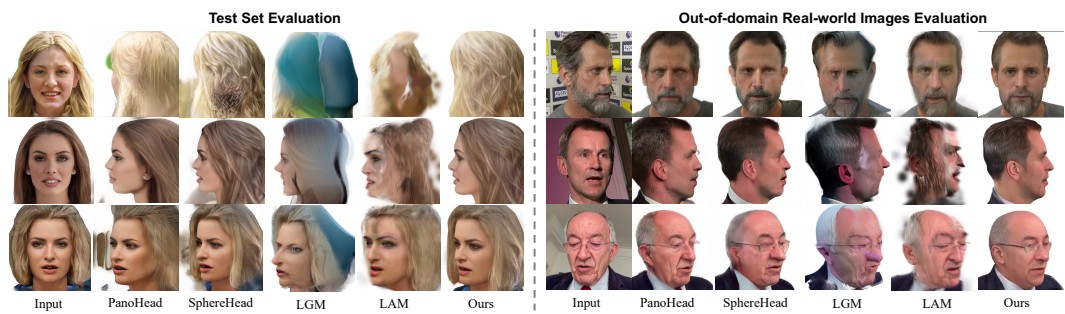

Figure 4: Visualization of reconstruction and novel view synthesis of different methods.

models. However, bad cases occur in the generated dataset, hurting the model training. Therefore, we remove bad cases from the sampled images manually, ending up with 48,135 subjects in Ours-Front and 117,186 subjects in Ours-360 datasets. More details of the dataset construction and labeling pipeline are in Sec E.

## 3.5 OPTIMIZATION AND REGULARIZATION

In the training phase, we randomly select $N_v$ different views of images of the same subject. We randomly choose one as the reference image for the Gaussian full-head reconstruction, while the others serve as novel view images for prediction. Note that both branches in our framework render images for supervision. We ensure the accuracy of the rendered RGB images from each branch by comparing them with the ground truth target images, utilizing a combination of $\mathcal{L}_1$ loss and perceptual loss for supervision:

$$\mathcal{L}_{rgb} = \lambda_1 \mathcal{L}_1 + \lambda_2 \mathcal{L}_{lpips}. \tag{7}$$

Additionally, for the sparse and dense Gaussian Splatting branch, we render the silhouette and supervise it using $\mathcal{L}_1$ loss, referred to as $\mathcal{L}_{mask}$. For the spherical triplane branch, we supervise the extracted feature with those in the pretrained SphereHead by $\mathcal{L}_1$ loss, denoted as $\mathcal{L}_{st}$. The total loss function is a weighted sum as:

$$\mathcal{L} = \lambda_3 \mathcal{L}_{rgb}^G + \lambda_4 \mathcal{L}_{rgb}^{ST} + \lambda_5 \mathcal{L}_{mask} + \lambda_6 \mathcal{L}_{st}, \tag{8}$$

where $\mathcal{L}_{rgb}^G$ denotes the RGB image loss for sparse and dense Gaussian Splatting and $\mathcal{L}_{rgb}^{ST}$ for the auxiliary spherical triplane branch, both denoted in Formula (7).

## 4 EXPERIMENTS

### 4.1 EXPERIMENTS SETTING

**Implementation Details.** Our framework is implemented in PyTorch, with the DINOv2 image feature extraction backbone frozen. The Transformer comprises 8 layers of basic blocks, featuring 16 attention heads and $C_t = 512$ feature dimensions. The extracted features are translated into Gaussian attributes through three multilayer perceptron (MLP) layers. The network is trained over 100 epochs using the Adam optimizer and a cosine warm-up scheduler. Hyperparameters are empirically set as $N_v = 6$, $\lambda_1 = \lambda_2 = 1.0$, $\lambda_3 = 1.0$, $\lambda_4 = 0.1$, $\lambda_5 = 1.0$, $\lambda_6 = 0.0001$, $\epsilon_{O_C} = 0.15$, $\epsilon_{O_D} = 0.056$, $M_D = 10K$, $K_m = 32$, $\lambda_{area} = 2.0e - 6$, $H_T = W_T = 64$.

**Datasets.** We employ the large-scale dataset that we have generated for training. To assess the 360-degree synthesis capability, we randomly select 100 subjects from our Ours-360 dataset for testing, and the remaining for training.

**Evaluation Metrics.** We evaluate the quality of our synthesized images using three quantitative metrics: Peak Signal-to-Noise Ratio (PSNR), Structural Similarity Index (SSIM), and Learned Perceptual Image Patch Similarity (LPIPS). To evaluate identity similarity (CSIM) of the reconstructed 3D assets, we compute the cosine distance of face recognition features (Deng et al., 2022). For assessing pose fidelity, we employ the Average Pose Distance (APD) (Deng et al., 2019).

## 4.2 MAIN RESULTS

**Qualitative Results.** Fig. 4 illustrates the novel view synthesis results after reconstruction from various methods on the test set and out-of-domain real-world images, highlighting the superior performance of PanoLAM. Unlike prior approaches, PanoLAM enhances texture detail, maintains identity fidelity, and ensures multiview consistency. Compared to PanoHead and SphreHead, there are no artifacts in the background as well. It is noteworthy that PanoLAM neither employs time-intensive test-time optimization (getting 800X speedup) nor relies on precise camera poses for accurate reconstruction, showcasing the efficacy of our pose-free framework.

**Quantitative Results.** We benchmark our framework and baselines on the test set. For each subject, we evaluate metrics on 4 viewpoints: the front, back, left, and right, except the CSIM metric only uses the front view. We use the default configurations for baselines. As in Table 2, PanoLAM demonstrates exceptional

Table 2: Quantitative evaluation on the testset.

| Method | PSNR↑ | LPIPS↓ | SSIM↑ | APD↓ | CSIM↑ |
|---|---|---|---|---|---|
| PanoHead-PTI | 16.260 | 0.165 | 0.704 | 0.029 | 0.680 |
| SphereHead-PTI | 16.983 | 0.142 | 0.698 | 0.018 | **0.808** |
| LGM | 11.472 | 0.444 | 0.572 | 0.102 | 0.457 |
| LAM | 15.627 | 0.245 | 0.647 | 0.056 | 0.767 |
| Ours | **23.494** | **0.107** | **0.793** | **0.015** | 0.790 |

reconstruction quality according to PSNR, SSIM, and LPIPS metrics. The CSIM metrics also indicate strong identity consistency of our methods. Remarkably, these achievements are coupled with fast reconstruction and rendering speeds, underscoring the effectiveness and efficiency of our approach.

Table 3: Running time for 3D avatar generation and rendering.

| Time | PH-PTI | SH-PTI | LAM | Ours |
|---|---|---|---|---|
| Recon.(s) | 89.4 | 96.2 | 1.4 | **0.11** |
| Render(ms) | 10.29 | 19.68 | **3.6** | **3.6** |

Table 4: Effect of Coarse-to-fine generation. OOM: Out of memory (80GB).

| Metrics | D-30K | C2F-30K | D-100K | C2F-100K |
|---|---|---|---|---|
| Infer.(s) ↓ | 0.49 | 0.082 | - | 0.11 |
| Memory(G)↓ | 17.1 | 7.22 | OOM | 12.7 |

Table 5: Module Ablations.

| Method | PSNR↑ | SSIM↑ |
|---|---|---|
| Full | 23.494 | 0.793 |
| w/o C2F | 22.312 | 0.731 |
| w/o ST ray agg. | 22.156 | 0.715 |
| 2K subj. | 19.391 | 0.699 |

**Reconstruction and Render Speed on Various Platforms.** Table 3 shows the comparative running time evaluation of all methods on an A100 GPU. Benefiting from our coarse-to-fine and feedforward framework design, our approach achieves a fast reconstruction speed. The Gaussian Splatting we chose as the 3D representation also enables fast rendering speeds compared to existing techniques.

**Analysis of Feature Sampling and Aggregation Mechanism from Spherical Triplane.** We analyze the feature aggregation mechanism from the spherical triplane of SphereHead to explore a better knowledge distillation strategy. In Fig. 5, we cast rays and visualize the aggregation weights of each sampled feature in the original ray marching. We plot 4 curves showing the weight values of 4 casting rays. Two phenomena occur as in the figure. Firstly, multiple peaks occur near the geometry surface hit by each ray, indicating that we need to sample and aggregate multiple points on a ray to fetch valuable features from the spherical triplane fully. Secondly, each weight curve appears to be a different weight function, indicating that we cannot apply the same manual weight functions, e.g., a Gaussian distribution, for each ray. Thereafter, we propose to leverage learnable MLPs to learn

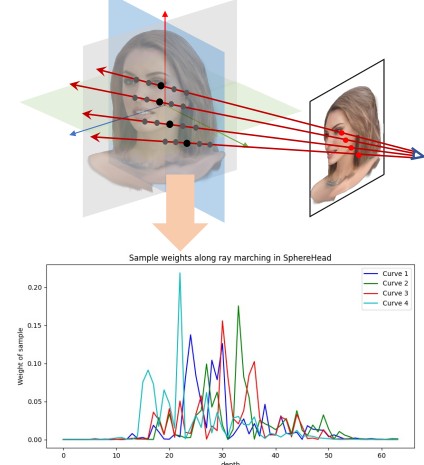

Figure 5: Analysis of ray marching aggregation weights in the original Spherical Triplane in SphereHead.

the feature aggregation function from large-scale data. Experimental results in the ablation study validate such a finding and the effectiveness of our design.

## 4.3 ABLATION STUDIES

**Effect of Coarse-to-fine Gaussian Point Generation.** Fig. 6 shows the effect of our coarse-to-fine Gaussian generation strategy. The sparse points are easier to optimize and can be easily deformed to carve the shape of the person, but they cannot model texture details like hair strands.

By densifying the points with topology prior in the FLAME mesh, our coarse-to-fine model can learn more detailed textures. In contrast, directly regressing the dense point cloud results in insufficient optimization of Gaussian points. Many points are not well optimized for deformation and remain on the FLAME surface, which are invisible and wasted, causing dual-layer geometry in the hair region as well. The quantitative results in Tab. 5 also support the effectiveness; without our coarse-to-fine training scheme (denoted as w/o C2F in the table), the performance drops. Moreover, as is shown in Table 4, our coarse-to-fine strategy saves computation and GPU memory consumption since fewer points are needed

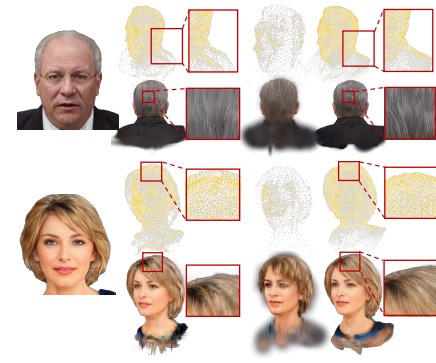

Input   W/o coarse-to-fine   Coarse   Fine (Ours)

Figure 6: Effect of coarse-to-fine generation.

for stacked cross-attention modules. The network forward time and training GPU consumption are also reduced with our coarse-to-fine training strategy.

**Effect of Different Dataset Scale.** Previous 3D head datasets lack subject diversity, as shown in Table 1. We ablate the effect of different numbers of subjects in Fig. 7 and Table 5 (2K subj.). The model trained on a limited number of subjects cannot generalize well to unseen subjects, causing performance drops visually and quantitatively. This shows the effectiveness of our proposed large-scale, diverse dataset for network generalizability.

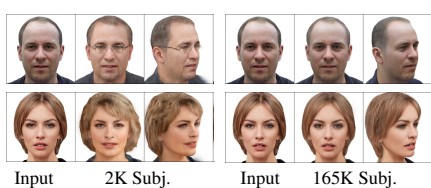

Input       2K Subj.       Input       165K Subj.

Figure 7: Comparison of our network trained on different numbers of subjects.

**Effect of Different Spherical Triplane Query Strategy.** In Fig. 8, we show the results of different query strategies from the spherical triplane. All strategies use the same feature renderer and frozen decoder from SphereHead for target image decoding. As shown in the figure, directly using the densified point cloud to query a single layer of features from the spherical tri-

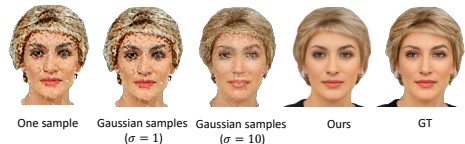

One sample   Gaussian samples   Gaussian samples   Ours   GT
             ($\sigma = 1$)       ($\sigma = 10$)

Figure 8: Comparisons on different spherical triplane feature query and aggregation strategies.

plane, as in (Zou et al., 2024), cannot fetch all valuable features, and the rendered images contain artifacts due to missing information. We then add 4 virtual cameras and cast rays from the camera center to each point and sample 32 layers of features near the point surface for feature aggregation. We set the margin between each sampled point to be the same as the fine ray marching stage in SphereHead. We try to assign manual weights sampled from a normal distribution centered at the selected point with standard deviation 1 and 10, and aggregate sampled features with a weighted sum. As is shown in the middle part of Fig. 8, the results look better but are still blurry. Our final solution utilizes learnable MLPs to learn the aggregation function for the NeRF-based rendering. As shown in the figure, ours gets the best results, showing the effectiveness of the proposed aggregation strategy. In Table 5, we also ablate the model without our proposed ray-based feature aggregation from spherical triplane (denoted w/o ST ray agg. in the table). The quantitative results drops and prove the effectiveness of the proposed module.

## 5 CONCLUSIONS

In this work, we present a novel feedforward large avatar model for Gaussian full-head synthesis from a single unposed image. To resolve the problem of lacking 3D assets for network training, we develop a large-scale, diverse 3D avatar dataset from trained 3D GANs. We also propose a coarse-to-fine mechanism with a network-free point cloud densification strategy for efficient reconstruction of a high-fidelity Gaussian full-head from a single image. We also introduce a dual-branch approach to distill knowledge from the spherical triplane prior and improve the reconstruction effectiveness. We analyze the feature sampling mechanism in spherical triplane and propose an efficient aggregation mechanism. Experimental results validate the effectiveness of our approach.

**Ethics Statement.** The generation of realistic avatars from images raises important ethical considerations, especially concerning privacy, consent, and the risks associated with the misuse of technology, such as the creation of deep fakes. It is essential to prioritize responsible use, ensuring adherence to ethical guidelines and standards to mitigate potential negative impacts.

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

## A  APPENDIX OVERVIEW

In this supplement, we present more results in Sec. B, more ablation studies in Sec. C, introduce our dataset generation and labeling process in Sec. E, discuss limitations of our work in Sec. D. Sec. F illustrates the usage of LLM.

## B  MORE RESULTS

In this section, we show more results as well as more comparisons with previous work.

**Quantitative results on out-of-domain real-world datasets.**  We conduct experiments on in-the-wild real-world data to evaluate the generalizability of the proposed methods. We sample 9 identities from the VFHQ Xie et al. (2022) test set and 9 identities from the Nersemble Kirschstein et al. (2023) dataset as GaussianAvatar Qian et al. (2024). We evaluate reconstruction quality using peak signal-to-noise ratio (PSNR) and structural similarity index (SSIM), following the protocol of GRAM-HD Xiang et al. (2023). Note that these

Table A1: Quantitative evaluation on real-world datasets.

| Method | PSNR↑ | SSIM↑ |
|---|---|---|
| PanoHead-PTI | 24.37 | 0.806 |
| SphereHead-PTI | 24.65 | 0.811 |
| LAM | 18.72 | 0.713 |
| Ours | **27.45** | **0.882** |

data are not seen during training, which is suitable to evaluate the out-of-domain reconstruction capability of different methods. For each method, we synthesize 30 views of the same subject, train the surface reconstruction method NeuS2 Wang et al. (2023b) on these images, and compute PSNR and SSIM on the resulting reconstructions. As shown in Table A1, PanoLam gets the best results on the reconstruction quality. While LAM He et al. (2025) is trained on videos with a limited viewing angle, it cannot reconstruct the full head well, especially in the side and back views, and the results degrade largely in the 360° evaluation. We also visualize some results in the right part of Fig. 4 and Fig. A4.

**360° synthesis results from our framework**  In Fig. A2, we render 360° images from our reconstructed Gaussian full-head given unposed images as input. As shown in the figure, after training on the proposed large-scale dataset, our framework can reconstruct Gaussian full-head and synthesize consistent novel view images.

**More comparison results with previous methods on the testsets.**  In Fig. A3, we show more results of different methods on the testset. Four views of images are rendered from the reconstructed 3D representation of different methods, showing the full-head quality. As shown in the figure, our framework gets much higher fidelity results and has fewer artifacts in unseen regions compared to previous works.

**More comparison results with previous methods on real-world images.**  In Fig. A4, we show more results of different methods on real-world images sampled from the VFHQ dataset. As shown in the figure, though trained on our synthesis only large-scale dataset, PanoLAM is able to generalize to the real-world images. Compared with previous methods, our framework reconstructs more texture details and maintains better identity fidelity.

## C  MORE ABLATION

**Effect of Different Number of Gaussian Points.**  We ablate the effect of different numbers of Gaussian points with our coarse-to-fine strategy in Fig. A1. As shown in the figure, a small number of Gaussian points cannot model texture details like hair strands and teeth, leading to blurry results in these regions. In contrast, more high-fidelity Gaussian avatars are reconstructed with an increasing number of Gaussian points. However, directly increasing the number of Gaussian points causes the problem of

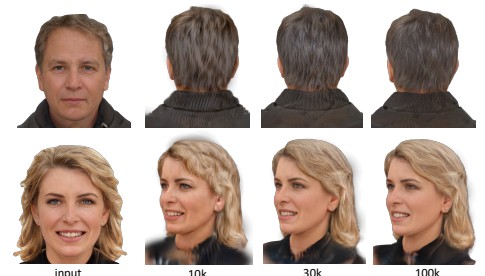

Figure A1: Comparison of different numbers of Gaussian points. Zoom in to see more details.

insufficient optimization of Gaussians and out-
of-memory issues. This further validate the effectiveness of our proposed coarse-to-fine training
strategy.

# D LIMITATIONS

Our dataset and model are built upon 3D GANs trained on different 2D datasets, e.g., FFHQ (Kazemi & Sullivan, 2014) for EG3D and the WildHead (Li et al., 2024), CelebA (Liu et al., 2015), FFHQ, LPFF (Wu et al., 2023), and K-Hairstyle (Kim et al., 2021) datasets for SphereHead. The proposed dataset is sampled from these trained 3D GANs, and the network trained on it has similar biases to these datasets. The sampled datasets contain less Asian data and no cartoon heads, leading to worse reconstruction results on Asian faces and bad results on cartoon faces from the proposed framework. A possible solution to this problem is to train the 3D GANs on a more diverse 2D dataset for more diverse 3D head avatar sampling and 3D avatar reconstruction network training. Our model only handles static full-head reconstruction, and the generated head is not animatable. We leave this as future work.

# E GENERATION AND LABELING PROCESS FOR OUR LARGE-SCALE SYNTHESIS DATASET.

In this section, we introduce our dataset generation process and labeling strategy.

**Dataset generation from trained 3D GANs.**   Given that EG3D is limited to rendering near-frontal images using its generated triplane NeRF, we sample images within a view range of approximately $72°$ from this model. Specifically, cameras are sampled from a pitch range of $\pm26°$ degrees and a yaw range of $\pm36°$ degrees relative to the front of the human face. To enhance the network's adaptability to real-world images captured by various cameras, we vary the camera radius and focal length. The camera radius is drawn from a normal distribution centered at 2.7 with a standard deviation of 0.1, while the focal length is sampled from a normal distribution centered at 18.83 with a standard deviation of 1. Each triplane generates 32 random images. For SphereHead (Li et al., 2024), we generate $360°$ dataset. Specifically, we initially sample 8 views evenly spaced along the equatorial plane to ensure full-head coverage, dividing the 360-degree range. Additionally, we sample 24 images at random angles and apply the same randomization of camera radius and focal length as used in the EG3D sampling process. Visualization of some cases is shown in Fig. A5.

**Dataset Labeling.**   Since the sampled data from 3D GANs may contain bad cases, this may hurt the training of the learning-based network. We also develop a labeling tool and remove bad cases manually to improve the dataset quality. Some examples of bad cases we removed are shown in Fig. A6.

# F STATE OF LLM USAGE

We use LLM to assist with paper polishing on grammar. Each sentence polished by LLM is checked to express our original meaning. There is no further use of LLM for the idea formulation, experiments, coding, and main paper writing.

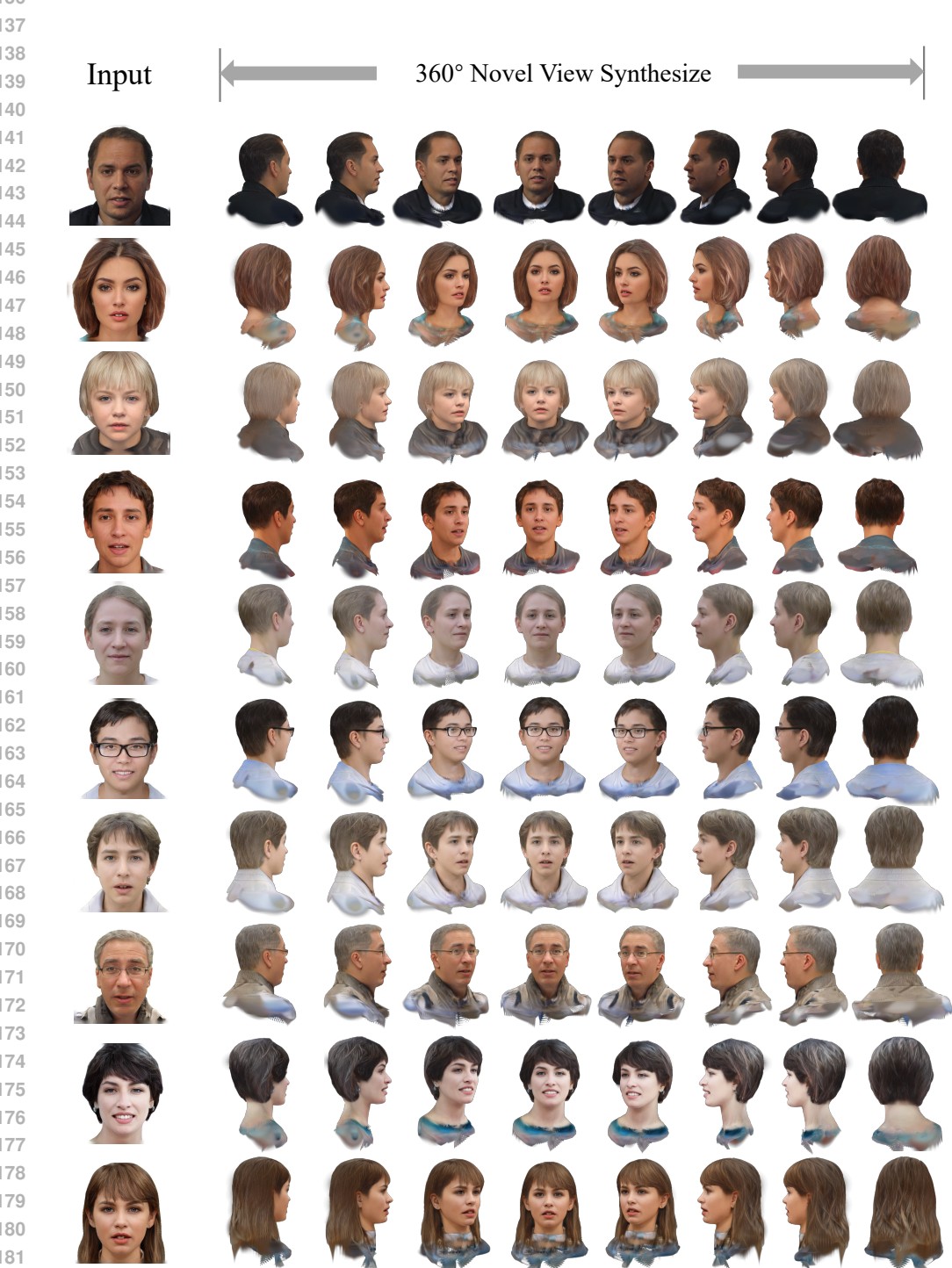

Figure A2: Visualization of our Gaussian full-head synthesis from one-shot unposed image.

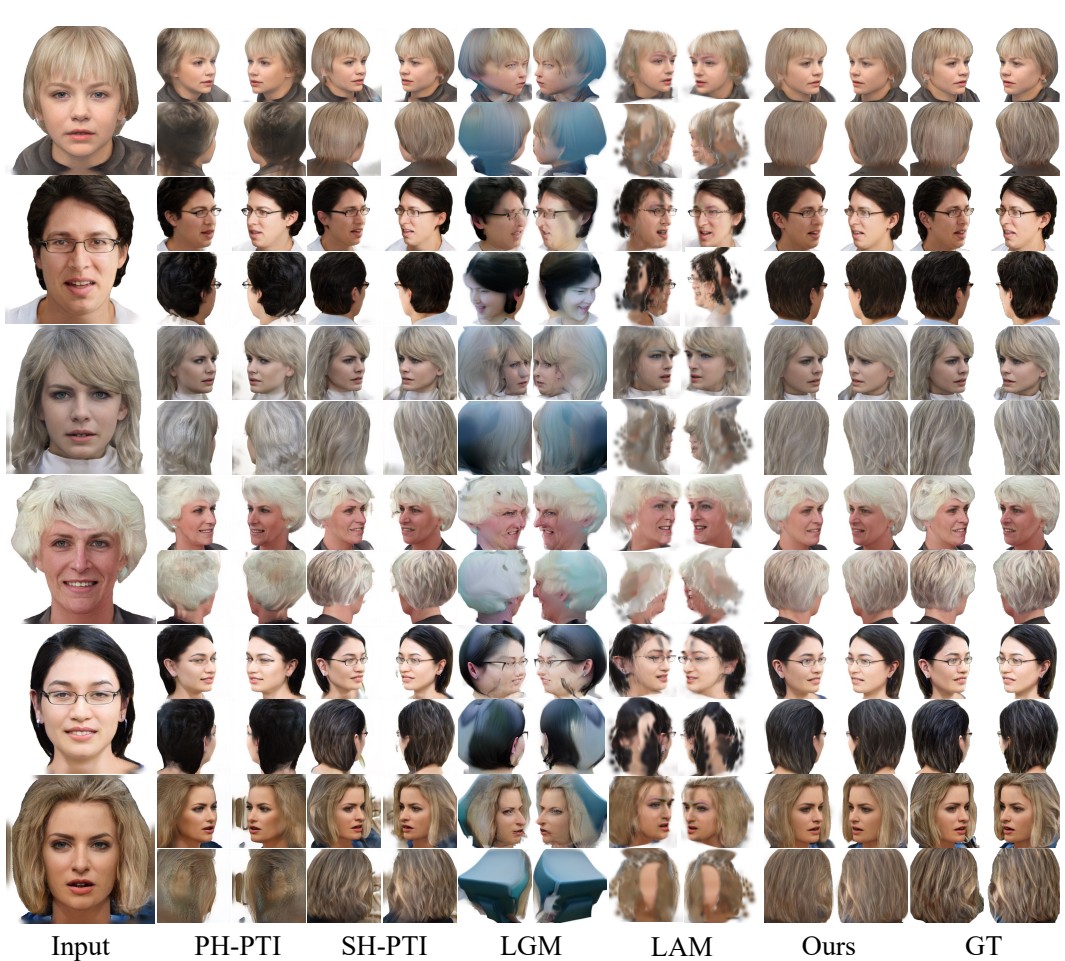

Input      PH-PTI      SH-PTI      LGM      LAM      Ours      GT

Figure A3: Comparison of reconstruction and novel view synthesis of different methods on the testset.

| Input | PH-PTI | SH-PTI | LGM | LAM | Ours |

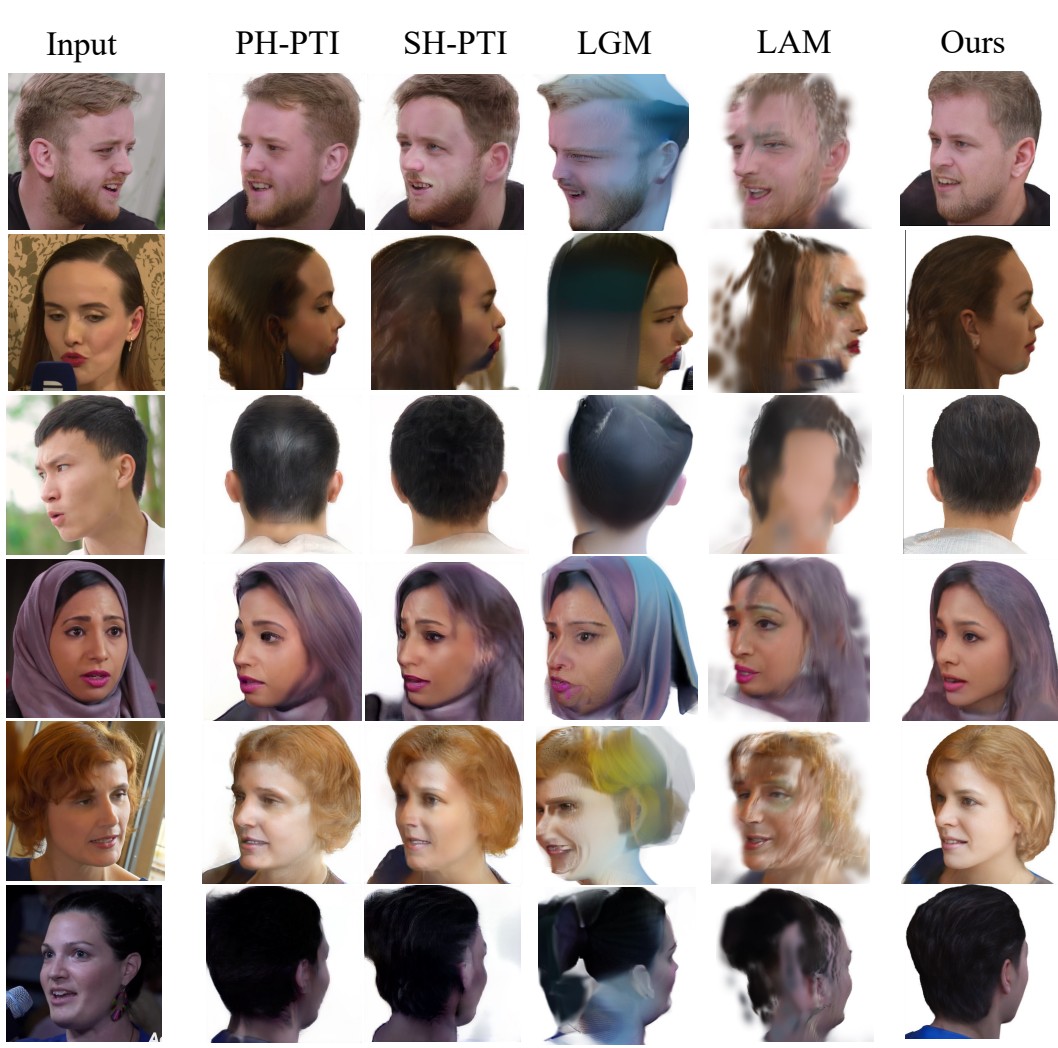

Figure A4: Comparison of reconstruction and novel view synthesis of different methods on in-the-wild real-world images.

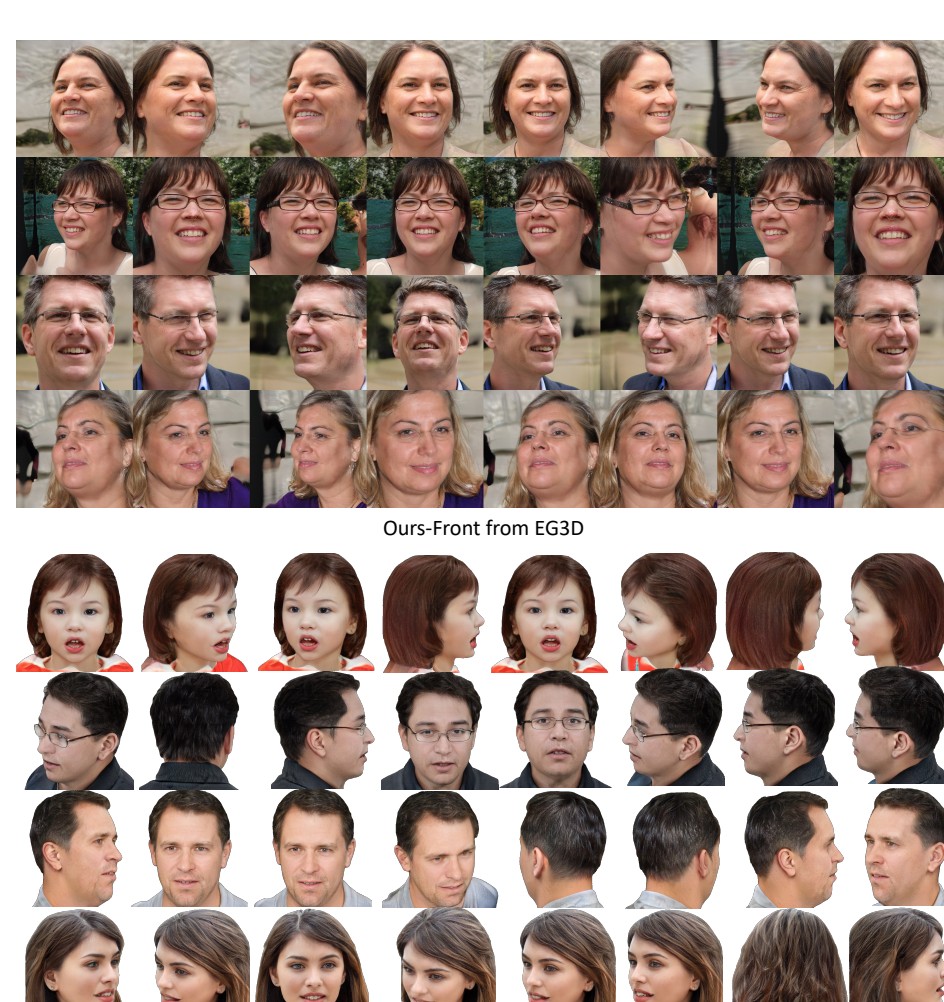

Ours-Front from EG3D

Ours-360 from SphereHead

Figure A5: Visualization of example cases in our sampled and cleaned datasets.

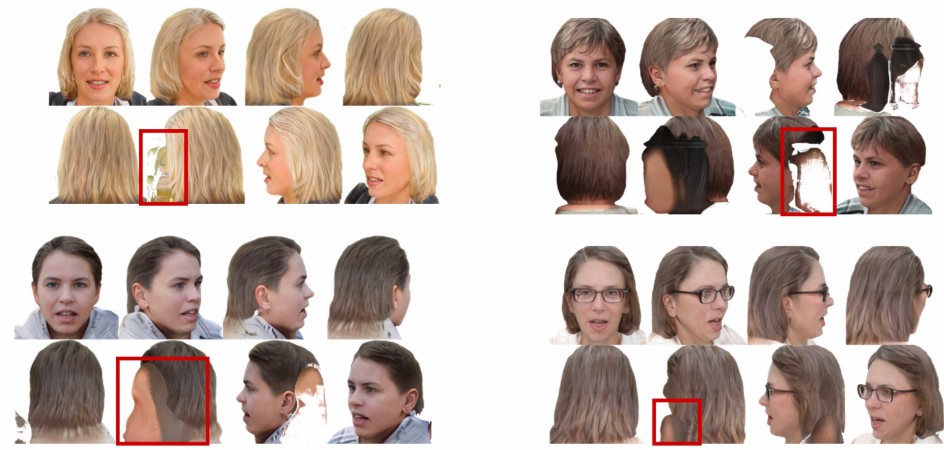

Figure A6: Examples of bad cases sampled from the 3D GANs prior, which are removed manually using our labeling tools.

