# OpenReview forum: "PanoLAM: Large Avatar Model for Gaussian Full-Head Synthesis from One-shot Unposed Image"
_ICLR.cc/2026/Conference — Submitted to ICLR 2026_

### Official Review · Reviewer_5G3n · 2025-10-25

**Soundness:** 2
**Presentation:** 3
**Contribution:** 2
**Rating:** 4
**Confidence:** 4

**Summary:**

This paper proposes a feed-forward framework for Gaussian full-head reconstruction from a single unposed image. The method is trained purely on a large-scale synthetic dataset generated from pretrained 3D GANs.  This paper proposes a coarse-to-fine Gaussian generation pipeline and a dual-branch design combining spherical triplane and point-based features to efficiently synthesis 3D head. Experiments demonstrate that the proposed framework outperforms existing approaches in both speed and quality.

**Strengths:**

1. Proposes a coarse-to-fine reconstruction framework that efficiently reconstructs a Gaussian full-head model.
2. Designs a dual-branch architecture that combines point-based representations with spherical triplane priors derived from pretrained 3D GANs, resulting in improved reconstruction performance.
3. Introduces a large-scale and diverse synthetic dataset to support 3D avatar reconstruction.

**Weaknesses:**

1. Lack of supplementary video: Since this is a head reconstruction method, the absence of supplementary videos makes it difficult to evaluate the consistency across different viewpoints and facial expressions.
2. Unclear handling of hair motion: The paper mentions that FLAME vertices cannot model long hair or accessories such as hats. However, the proposed method also appears to use FLAME’s original LBS for animation. As FLAME’s LBS does not model hair dynamics, it remains unclear how well the proposed Gaussian head handles hair motion during animation.
3. Limited discussion on dataset influence: In the ablation study (Table 5), the dataset scale significantly affects the model’s performance. The authors should provide more detailed discussions comparing their training dataset with those used in other state-of-the-art methods.

**Questions:**

1. Will you release your dataset?

**Details Of Ethics Concerns:**

The proposed method enables high-fidelity 3D head reconstruction from a single image, which raises potential ethical concerns regarding privacy, identity misuse, and deepfake generation. Since realistic 3D heads can be synthesized without subject consent, the technique could be misused to create unauthorized virtual identities or manipulated facial animations, leading to privacy violations or reputational harm.

---

### Official Review · Reviewer_9eHv · 2025-11-01

**Soundness:** 2
**Presentation:** 3
**Contribution:** 2
**Rating:** 4
**Confidence:** 4

**Summary:**

The paper addresses the task or reconstruction a static 3DGS representation from a single head image.

Similar to Large-Reconstruction-Models (LRM), the authors propose to perform cross attention from learnable queries to DINOv2 tokens. The authors leverge two types of queries. The first set of queries is anchored to the vertices of the FLAME mesh, which can be efficiently be upsampled by barycentric interpolation using the FLAME topology. The second set of queries stems from triplanes, which also aggreagte features from the DINOv2 Tokens. The resulting features from set 1 then extract information from the triplane features using samples along 4 rays to a set of 4 fixed cameras. Finally, an MLP learns to map the final features into Gaussian attributes.

Next to standard rendering losses during training, the authors propose to distill infromation from SphereHead, by enforcing that the resulting triplane features can be decoded into RGB using the frozen SphereHead decoder.

Importantly, the method is trained on purely synthetic data generated from SphereHead and PanoHead. Not surprisingly, the baseline methods, which have been trained on completely different datasets, perform worse on 100 held-out identities.

**Strengths:**

- The main strength of the paper is its quality, and usability. Reconstructing 3D heads (although they are non-animatable) is a relavant task, and having a feed-forward method for it is certainly more useful than running costly comparatively PTI. The quantitative numbers support the proposed method.
- Thanks to the 3DGS representation, rendering is fast, and 3D consistent. The latter is e.g. evaluated in the supplemental, where the proposed method outperforms all baselines.
- The paper is well-written and easy to follow.

**Weaknesses:**

- While the authors cite LivePortrait3D and TriplaneNet, they do not consider such works as baselines, although they follow the exact same motivation as the propsed method. Therefore, I believe that such method consititue a crucial baselines, and should be compared against using the same training data. This would validate the proposed approach to learn a single-image2-3D-predictor. Due to the existence of such previous work, the novelity is limited on a high leve. However, some technical novelty remains, which should be highlight better experimentally, see below.
- The results on the real world examples in Fig. 4 show that PanoLAM heavily suffers from identitiy drift. I assume that this is caused by the fact that real images are out-of-distribution for the generated data. In my opinion the paper should include reconstruction evaluation (not just view consistency) on a multi-view dataset such as NeRSemble. The task would be simple: use 1 of the 16 cameras as input, evaluate PSNR, SSIM and LPIPS on the other 15 views.
- The proposed evaluation heavily favors the proposed method, since 100 subjects from the  exact same training data distribution are used for evaluation. Either baselines, like LAM,  should be trained/fine-tuned on the same data, or other datasets (see previous point) should be used for evaluation. As such, beating the baselines is not very surprising.
-

**Questions:**

- What exactly is the benefit of the proposed upsampling approach? Linear interpolation seems quite simple and cannot come up with new information. What about pixel-shuffle upsampling, as done e.g. in GRM, LVSM? Is it mainly about training stability?
- Overall, the pipeline seems to be rather convoluted. Two sets of query tokens are used to attent to DINOv2 features. And then for resulting positions of the frist set of queries, the information from the second set of queries is extracted. What is the intuiton behind this? At least accroding to my intuition, only using point queries should work fairly well, as e.g. demonstrated in LAM. (yes LAM cannot generate complete heads, but this is caused by its training data).
- What is the effect of using a frozen SphereHead decoder and additional loss? This forces the resulting triplane features to behave similar to what SphereHead does, or is there more to it? Any effect on the training stability, convergence speed etc.?
- How is the ground truth for $\mathcal{L}_{st}$ in eq. (8) obtained?
- Why is SphereHead obtainined such low numbers in Table 2? After all it was used to construct part of the dataset, and should give reasonable PTI results, e.g. as shown in Figure 4. However the numbers look very bad? Does this e.g. have to do with inproper background segmentation?

---

### Official Review · Reviewer_4ybf · 2025-11-01

**Soundness:** 3
**Presentation:** 3
**Contribution:** 3
**Rating:** 6
**Confidence:** 5

**Summary:**

In this paper, a method for turning a single, unposed RGB image of a human head into a Gaussian head portrait is proposed. The method is based on a neural pipeline with several pathways: the image features are obtained by DINOv2 and merged via cross-attention with point queries coming from the FLAME parametric model, as well as used to condition a network branch that originates from spherical triplanes and produces triplane features. The results of merging image features and point queries are densified due to the limited precision of FLAME and used to produce the Gaussians. Input spherical triplanes are used to distill prior knowledge from a 3D GAN. Due to the pipeline being direct and feedforward, the method is fast and easily scalable, which presents a considerable advantage in practice. The results are demonstrated on the benchmarks of image quality, consistency with the ground truth, inference time measurements.

**Strengths:**

- Very good results.
- The method is simple yet efficient. The network is feedforward and fast. The proposed approach is much more efficient than many available approaches that depend on diffusion models and obtain on par or worse results; also, it is more efficient than GAN inversion approaches. Coarse+fine refinement makes sense given a small number of points (5K) in FLAME. The generated portrait is Gaussian, which is much more reliable and easier to use than 2D generations from EG3D-like networks; the training is also simpler, as it does not require adversarial losses.
- Very thorough and trustworthy experimental section with many comparisons and ablations.

**Weaknesses:**

- Even though there are visuals from various views shown in Appendix, there are no supplementary video to inspect the quality of the Gaussian reconstruction from different sides better.
- The method is still relatively limited to the EG3D-style coordinate system, which means it would require a specific focal length and cropping during inference. This should limit the use of the method in applications. However, I could imagine this method retrained on different datasets such as Ava-256 or Nersemble (the only caveat for this would be spherical triplanes coming from SphereHead, since for them to be calculated correctly, the input image must still be in the EG3D-style coordinate system). There are some out-of-domain results shown in the Appendix, but, e.g., no visuals are shown on Nersemble; it's unclear how it would perform e.g., on more wide-angle (selfie-style) or large focal length images.
- In Table 1, the proposed Ours-Front and Ours-360 datasets are pitched from the standpoint of avatar datasets. For this to be true, 3D-consistency of the generations from different views (in the geometric & photometric sense) has to be measured, as well as completeness of the generation. Also, the demographics might be much less diverse than in real datasets, as well as the resolution, lighting variation, and many other characteristics. In other words, I believe pitching the synthetically generated dataset in the same table as real datasets can be misleading for the reader.
- In the method figure, triplanes are spherical triplanes but shown as vanilla triplanes.
- A natural application of this method would be "live portraits" -- generating Gaussians in a temporal setting from e.g. a webcam stream. This would likely be a real-world application of this method. I think it is very important to show that; also, without such a test, it is hard to evaluate the temporal consistency of generations. Such results are shown e.g. in LAM's supplementary video.

**Questions:**

- The inference time in the table: does it include preparing the input triplanes?

---

### Official Review · Reviewer_bNRW · 2025-11-05

**Soundness:** 3
**Presentation:** 3
**Contribution:** 2
**Rating:** 4
**Confidence:** 3

**Summary:**

This paper proposes a method full head reconstruction from only a single image. The method follows a coarse-to-fine 3D gaussian splatting strategy with additional supervision via SphereHead, a triplane based head reconstruction method. First, a coarse 3D gaussian splat is initialized via the FLAME mesh. This initial splat is refined by conditioning on DinoV2 features of the input image via cross attention to generate per-point deformations, color and gaussian parameters. The coarse mesh is then subdivided and refined by conditioning on the corase mesh itself and features aggregated from a spherical triplane to regress the ground truth. Quantitatively the method outperforms prior work, however, qualitatively I see major identity shifts.

**Strengths:**

1) While no individual part of the paper is novel, the whole system is
2) Distillation of knowledge into 3DGS is a good idea, though I'm afraid it is not well ablated.
3) Strong quantitative results.

**Weaknesses:**

1) The biggest weakness is the identity shifting in real images, often the reconstruction does not look like the person (see Fig 4)
2) The need for SphereHead is not very well justified.

**Questions:**

While I suspect train/test mismatch explains the identity shift, it would be great to test this and be given an explanation by the authos.

---

### Meta-Review · Area_Chair_UXqt · 2026-01-07

**Summary:**

This paper proposes PanoLAM, a feed-forward pipeline to reconstruct a full-head 3D Gaussian Splatting avatar from a single unposed image, trained purely on a large synthetic dataset generated from pretrained 3D GANs (with a coarse-to-fine FLAME-initialized point branch and a spherical-triplane prior/distillation branch).

Reviewers agree the system is efficient and reports strong in-distribution quantitative results, but raise concerns about identity fidelity and evaluation/generalization.

With three reviewers below threshold and no rebuttal to clarify or add evidence, I recommend rejection.

**Reviewer Concerns:**

No author rebuttal was provided, so concerns remain as stated in the initial reviews.

1. Identity drift on real images: multiple reviewers report noticeable identity shifts on real-world examples (e.g., Fig. 4), undermining practical one-shot reconstruction claims.

2. Evaluation fairness / generalization: main benchmarking uses held-out identities from the same synthetic generation distribution; reviewers request baselines retrained/fine-tuned on the same data and/or reconstruction evaluation on real multi-view datasets (e.g., NeRSemble) to validate out-of-distribution performance.

3. Novelty / missing baselines: novelty is viewed as limited at a high level, and reviewers suggest including stronger, closely related baselines (e.g., triplane-based single-image predictors with similar motivation).

4. Justification/ablation gaps: the necessity/effect of the SphereHead distillation/triplane branch is not fully justified to skeptics; inference-time accounting is unclear (e.g., whether triplane preparation is included).

**Reviewer Scores:**

bNRW: 4

4ybf: 6

9eHv: 4

5G3n: 4

---

### Decision · Program_Chairs · 2026-01-26

Reject